A contrast-enhanced CT-based whole-spleen radiomics signature for early prediction of oxaliplatin-related thrombocytopenia in patients with gastrointestinal malignancies: a retrospective study

Dai Yuhong 1
Cheng Yiqi 2
Zhou Ziling 2
Li Zhen 2
Luo Yan yanluo@hust.edu.cn 2
Qiu Hong qiuhong@hust.edu.cn 1
1 Department of Oncology, Tongji hospital, Tongji Medical College, Huazhong University of Science and Technology , Wuhan , China
2 Department of Radiology, Tongji hospital, Tongji Medical College, Huazhong University of Science and Technology , Wuhan , China
Güven Deniz Can
Electronic publication date: 2023 Oct 13
Publication date: 2023
Volume: 11
Electronic Location ID: e16230
Received 2023 Mar 7; Accepted 2023 Sep 12
Copyright: ©2023 Dai et al.
Copyright year: 2023
Copyright holder: Dai et al.
License: This is an open access article distributed under the terms of the Creative Commons Attribution License, which permits unrestricted use, distribution, reproduction and adaptation in any medium and for any purpose provided that it is properly attributed. For attribution, the original author(s), title, publication source (PeerJ) and either DOI or URL of the article must be cited.
License URL: https://creativecommons.org/licenses/by/4.0/

Keywords: Radiomics, Oxaliplatin-based chemotherapy, Thrombocytopenia, Spleen, Computed tomography, Prediction model

Funding: The National Natural Science Foundation of China (NSFC) 82001786 The Top Independent Innovation Physician Funded Projects of Huazhong University of Science and Technology 3011540024 5001540074 5001540095 The Wuhan Young and Middle-age Medical Backbone Training Project 2016whzqnyxggrcl This work was supported by grants from the National Natural Science Foundation of China (NSFC) No. 82001786, the Top Independent Innovation Physician Funded Projects of Huazhong University of Science and Technology (No. 3011540024, 5001540074, 5001540095) and the Wuhan Young and Middle-age Medical Backbone Training Project (No. 2016whzqnyxggrcl). The funders had no role in study design, data collection and analysis, decision to publish, or preparation of the manuscript.

==============================
Background

Thrombocytopenia is a common adverse event of oxaliplatin-based chemotherapy. Grade 2 or higher oxaliplatin-related thrombocytopenia may result in dose reduction, discontinuation or delay initiation of chemotherapy and may adversely affect the therapeutic efficacy and even overall survival of patients. Early recognition of patients at risk of developing grade 2 or higher thrombocytopenia is critical. However, to date there is no well-established method to early identify patients at high risk. The aims of this study were to develop and validate a contrast-enhanced CT-based whole-spleen radiomics signature for early prediction of grade 2 or higher thrombocytopenia in patients with gastrointestinal malignancies treated with oxaliplatin-based chemotherapy and to explore the incremental value of combining the radiomics signature and conventional clinical factors for risk prediction.

Methods

A total of 119 patients with gastrointestinal malignancies receiving oxaliplatin-based chemotherapy from March 2017 to December 2020 were retrospectively included and randomly divided into a training cohort (n = 85) and a validation cohort (n = 34). Grade 2 or higher thrombocytopenia occurred in 26.1% of patients (22 and nine patients in the training and validation cohort, respectively) with a median time interval of 101 days from the start of chemotherapy. The whole-spleen radiomics features were extracted on the portal venous phase of the first follow-up CT images. The least absolute shrinkage and selection operator (LASSO) algorithm was applied to select radiomics features and to build the radiomics signature for the prediction of grade 2 or higher thrombocytopenia. A clinical model that included clinical factors only and a clinical-radiomics model that incorporated clinical factors and radiomics signature were constructed. The performances of both models were evaluated and compared in the training, validation and the whole cohorts.

Results

The radiomics signature yielded favorable performance in predicting grade 2 or higher thrombocytopenia, with the area under the curve (AUC), sensitivity and specificity being 0.865, 81.8% and 84.1% in the training cohort and 0.747, 77.8% and 80.0% in the validation cohort. The AUCs of the clinical-radiomics model in the training and validation cohorts reached 0.913 (95% CI [0.720–0.935]) and 0.867 (95% CI [0.727–1.000]), greater than the AUCs of the clinical model. Integrated discrimination improvement (IDI) index showed that incorporating radiomic signature into conventional clinical factors significantly improved the predictive accuracy by 17.0% (95% CI [4.9%–29.1%], p = 0.006) in the whole cohort.

Conclusions

Contrast-enhanced CT-based whole-spleen radiomics signature might serve as an early predictor for grade 2 or higher thrombocytopenia during oxaliplatin-based chemotherapy in patients with gastrointestinal malignancies and provide incremental value over conventional clinical factors.

Introduction

Oxaliplatin, as a third-generation platinum-based drug, inhibits DNA replication and transcription through the formation of cross-double-stranded DNA, inducing cell death, thus plays an anti-tumor role (Jardim et al., 2012). Since its approval for the treatment of metastatic colorectal cancer in 2002, oxaliplatin has been widely used in gastrointestinal malignancies, such as gastric cancer, colorectal cancer and pancreatic cancer (Jardim et al., 2012).

Thrombocytopenia is a common adverse event of oxaliplatin-based chemotherapy with reported incidence ranging 45%–77% in colorectal cancer (Grothey et al., 2018; Colucci et al., 2005; Andre et al., 2004). Although the incidence of severe thrombocytopenia is relatively low, this adverse event may result in dose reduction and discontinuation of treatment that ultimately may affect the therapeutic efficacy and even overall survival of patients (Jardim et al., 2012). Furthermore, oxaliplatin is increasingly used in neoadjuvant and conversion chemotherapy before surgery in gastrointestinal malignancies. In this context, oxaliplatin-induced thrombocytopenia may increase risks of bleeding and perioperative complications, leading to higher blood transfusion rates and longer hospital stays (Jardim et al., 2012). Early recognition of patients at risk of developing thrombocytopenia is of great clinical importance to ensure close monitoring and prompt intervention. Unfortunately, to date there is no well-established method to early identify patients at high risk of oxaliplatin-related thrombocytopenia.

Previous studies showed that splenomegaly, as observed on follow-up CT images was correlated with the reduction in platelet count in patients receiving oxaliplatin-based chemotherapy (Angitapalli et al., 2009; Overman et al., 2010; Kim et al., 2016) and might be recognized as a potential etiology for oxaliplatin-related persistent thrombocytopenia (Overman et al., 2010).

Radiomics, which extracts high-throughput quantitative feature data otherwise invisible to the naked eye from digital medical images, is now gaining importance in medical research (Lambin et al., 2017). Radiomics signatures based on contrast-enhanced CT have been applied in the diagnosis and prognosis of various tumors with promising results, especially in the early prediction of treatment response, survival and recurrence (Trebeschi et al., 2019; Dohan et al., 2020; Park et al., 2020; Ji et al., 2020).

Based on the possible role spleen play on the oxaliplatin-related thrombocytopenia and the encouraging predictive ability of radiomics, we hypothesized that spleen radiomics signature based on the follow-up CT might be able to predict oxaliplatin-related thrombocytopenia. Therefore, this study aims to develop and validate a contrast-enhanced CT-based whole-spleen radiomics signature for early prediction of oxaliplatin-related thrombocytopenia in patients with gastrointestinal malignancies and to explore the incremental value of combining the radiomics signature and conventional clinical factors for risk prediction of thrombocytopenia.

Materials & Methods

Patient population

This retrospective observational study was approved by our institutional review board (approval number: TJ-IRB20201221) and informed patient consent was waived. Patients with pathologically or cytologically diagnosed gastrointestinal malignancy who were admitted to the Cancer Center of Tongji Hospital, Tongji Medical College, Huazhong University of Science and Technology from March 2017 to December 2020 were reviewed. Patients were included if they were over 18 years of age and chemotherapy naive, received a minimum of two cycles and a maximum of twelve cycles of oxaliplatin-based chemotherapy (either first-line or adjuvant, including modified FOLFOX6 (mFOLFOX6), CAPOX and SOX regimens) and underwent follow-up contrast-enhanced abdominal CT within three months after the start of chemotherapy. Patients with baseline platelet count <100*109/L or received concurrent radiotherapy were excluded. Oxaliplatin-based chemotherapy regimens were listed in Table S1. The final study population consisted of 119 patients. Patients were randomly divided into the training (n = 85) and validation (n = 34) cohort with a ratio of 7:3.

Patients were followed up regularly and laboratory tests, including complete blood count were monitored before and during each cycle of chemotherapy. Thrombocytopenia during chemotherapy was recorded and was graded as per National Cancer Institute’s Common Terminology Criteria for Adverse Events (CTCAE; version 5.0) based on platelet count as follows: grade1, ≥75*109/L and <100*109/L; grade 2, ≥50*109/L and <75*109/L; grade 3, ≥25*109/L and <50*109/L; grade 4, <25*109/L (National Cancer Institute, 2017). According to CTCAE, no intervention is indicated for grade 1 thrombocytopenia. Therefore, in this study we focused on grade 2 or higher thrombocytopenia, since more intensive medical observation and intervention are required in these cases. Patients were divided into two groups, the thrombocytopenia group (TCP Group) and the non-thrombocytopenia group (Non-TCP Group), based on whether they developed grade 2 or higher thrombocytopenia during chemotherapy. Follow-up contrast-enhanced abdominal CT were performed every 1.5–2 months after the start of chemotherapy. Demographic, clinical, laboratory and imaging data of participants were recorded. The workflow of this study was shown in Fig. 1.

Figure 1 The workflow of this study.

LASSO: least absolute shrinkage and selection operator.

CT image acquisition

All contrast-enhanced abdominal CT scans were acquired on a 64-slice CT system (Discovery C750 HD; GE Healthcare). The following parameters were used: tube voltage, 100–120 kV; tube current, 200–350 mA; slice thickness, 5 mm; matrix, 512*512. Non-ionic contrast medium Iopromide (Ultravist, Bayer Healthcare, Wayne, NJ, iodine concentration, 370 mg/mL) was injected intravenously at a rate of 3 ml/s followed by a 25 ml saline flush. Bolus tracking technique was used to automatically trigger arterial phase, portal venous phase and delayed phase acquisition 5–8 s, 20–25 s and 180–240 s after the attenuation of abdominal aorta reached 150 HU, respectively. The raw data were reconstructed into 1.25 mm slice thickness and exported in DICOM format for analysis.

Spleen segmentation, volume measurement and radiomics feature extraction

3D Slicer (version 4.11.20210226, http://www.slicer.org), an open-source software was used to segment spleen, measure volume and extract radiomics features from the first follow-up contrast-enhanced abdominal CT images. A radiologist with 5 years of experience in abdominal imaging semi-automatically segmented the whole spleen on portal venous phase images using the Nvidia AI-Assisted Annotation (Nvidia AIAA) extension package implemented in 3D Slicer. After several boundary points had been set near the edge of spleen on axial, sagittal and coronal views, 3D Slicer software segmented the whole spleen within a minute, the radiologist then interactively corrected the segmentation if necessary and large blood vessels and lesions were carefully excluded. An example of the semi-automatic whole spleen segmentation was showed in Fig. S1. All segmentations were further double-checked by another radiologist with 10 years of experience in abdominal imaging. The volume of spleen was calculated after semi-automatic segmentation. Finally, a total of 851 radiomics features, consisted of first-order features (n = 18), shape features (n = 14), second- and higher-order features (n = 75) and wavelet features (n = 744) were extracted using the SlicerRadiomics extension package implemented in 3D Slicer. To determine the intra- and inter-reader reproducibility of radiomics features, 20 randomly-selected cases were segmented by the same radiologist again after a period of 1 month and by another radiologist with 10 years of experience in abdominal imaging.

Radiomics feature selection and radiomics signature development

Before feature selection, Z-score normalization was performed to rescale all radiomics features in a standard normal distribution in the training cohort. The mean and standard deviation of each radiomics feature in the training cohort were used for the normalization in the validation cohort. Radiomics features with intraclass correlation coefficients > 0.75 in intra- and inter-reader reproducibility tests were considered reproducible and included in feature selection. The independent two-sample t-test and the Mann–Whitney U test were then conducted to identify statistically significant radiomics features between the TCP Group and the Non-TCP group in the training cohort. The least absolute shrinkage and selection operator (LASSO) logistic regression algorithm was applied to further select features strongly associated with the development of grade 2 or higher thrombocytopenia in the training cohort. LASSO is a simple dimensionality reduction method with smaller mean squared error and deals with multicollinearity problems and overall variable selection, therefore is widely used in a variety of model fitting (Lee et al., 2014). The glmnet package in R software (version 3.6.1; R Core Team, 2019) was used for LASSO regression with penalty parameter tuning conducted by 10-fold cross-validation. A Radiomics signature, radiomics score (Rad-score) was constructed by a linear combination of the selected features weighted by their respective LASSO coefficients. The diagnostic performance of radiomics signature was evaluated through receiver operating characteristic (ROC) curve analysis in the training and validation cohorts. The robustness and stability of radiomics signature was evaluated by repeated 10-fold cross validation (200 times) in the whole cohort.

Clinical model and clinical-radiomics model development and performance evaluation

Clinical factors including age, gender, body mass index (BMI), primary tumor site, liver metastases, cumulative dose of oxaliplatin, number of oxaliplatin-based chemotherapy cycles, oxaliplatin-based chemotherapy regimen, concurrent bevacizumab treatment and spleen volume were analyzed using univariable logistic regression in the training cohort. Clinical factors with p < 0.05 were included into multivariable logistic regression to build models predicting the development of grade 2 or higher thrombocytopenia in the training cohort. A clinical model that included only clinical factors, as well as a clinical-radiomics model that incorporated clinical factors and radiomics signature were built. Multicollinearity was assessed using variance inflation factor (VIF) before multivariable logistic regression. A nomogram of the clinical-radiomics model was constructed to provide a visual, intuitive and individualized prediction of the development of grade 2 or higher thrombocytopenia during oxaliplatin-based chemotherapy.

Model performances including discrimination and calibration of both clinical and clinical-radiomics models were evaluated to determine the incremental value of radiomics signature over conventional clinical factors. The area under the ROC curve was calculated to assess the discrimination of models. The bootstrap method was performed to compare the area under the curve (AUC) of both models. The Hosmer–Lemeshow goodness-of-fit test and the Akaike information criterion (AIC) were used to assess the calibration of the models (Steyerberg et al., 2010). A non-significant p value of the Hosmer–Lemeshow test indicates good calibration. The AIC is a statistic evaluating the goodness-of-fit as well as the simplicity of a prediction model. A lower AIC value indicates a better-fit model. Empirically, if the AIC difference between models (ΔAIC) > 2, the model with the lower AIC value is considered substantially better fitted (Hu et al., 2021). Integrated discrimination improvement (IDI) index was calculated to evaluate the incremental prognostic accuracy after adding radiomics signature over conventional clinical factors for risk prediction of thrombocytopenia (Pencina et al., 2008).

Statistical analysis

Statistical analysis was performed with R software (version 3.6.1; R Core Team, 2019). The Chi-square test or Fisher’s exact test were used to compare categorical variables while the independent two-sample t-test and the Mann–Whitney U test were used to compare continuous variables between the training and validation cohorts. Two-tailed p < 0.05 was considered statistically significant.

Results

Patient characteristics

A total of 119 patients (70 men; median age, 54 years; interquartile range, 50–64 years) with gastrointestinal malignancy, including 74 patients with colorectal cancer and 45 patients with gastric cancer were included into this study. Before chemotherapy 24 patients (20.2%) presented with synchronous liver metastases. The mean BMI of all patients was 21.50 ± 2.65 kg/m2. All patients received at least two cycles of oxaliplatin-based chemotherapy (range, 2–12 cycles; median number of cycles, 6). The median cumulative dose of oxaliplatin was 780 mg/m2 (range, 255–1,040 mg/m2). The majority of patients (75 out of 119, 63.0%) was treated with CAPOX regimen. About 10.1% of patients received bevacizumab in combination with CAPOX (7.5mg/kg every 3 weeks, n = 4) or mFOLFOX6 (5 mg/kg every 2 weeks, n = 8) regimen. The baseline characteristics of patients in the training and validation cohorts are summarized in Table 1. No significant difference was found between these two cohorts.

Table 1 Patient clinical characteristics in the training and validation cohorts.

Characteristics	Training cohort
(n = 85)	Validation cohort
(n = 34)	p value	
Age(years)	54.05 ± 10.01	54.94 ± 9.94	0.660	
Gender	
Male	52[61.2]	18[52.9]	0.418	
Female	33[38.8]	16[47.1]		
BMI (kg/m2)	21.47 ± 2.52	21.58 ± 2.98	0.836	
Primary tumor site			0.188	
Colorectal cancer	56[65.9]	18[52.9]		
Gastric cancer	29[34.1]	16[47.1]		
Liver metastases			0.051	
Yes	21[24.7]	3[8.8]		
No	64[75.3]	31[91.2]		
Chemotherapy regimen			0.075	
CAPOX	50[58.8]	25[73.5]		
mFOLFOX6	27[31.8]	4[11.8]		
SOX	8[9.4]	5[14.7]		
Cumulative dose of oxaliplatin (mg/m2)	780(650,841)	769(544,780)	0.454	
Number of chemotherapy cycles	6(5,8)	6(5,6.25)	0.086	
Bevacizumab			0.505	
Yes	10[11.8]	2[5.9]		
No	75[88.2]	32[94.1]		
Grade 2 or higher thrombocytopenia			0.947	
Yes	22[25.9]	9[26.5]		
No	63[74.1]	25[73.5]		
Spleen volume (cm3)	218.40(175.31,278.12)	204.82(164.94,254.01)	0.459	
Notes.

Numbers in square brackets are percentages and numbers in parentheses are interquartile range.

Continuous variables are presented as mean ± standard deviation or median (interquartile range).

BMI body mass index

Follow-up

Thrombocytopenia was observed in 68 out of the 119 patients (57.1%) during oxaliplatin-based chemotherapy. Slightly more than half of the patients who developed thrombocytopenia (37 out of 68, 54.4%) were mild cases (grade 1) and no medical intervention was required. Grade 2 or higher thrombocytopenia occurred in 31 patients (26.1%), with a median time interval of 101 days (interquartile range, 70–141 days) from the start of chemotherapy to the onset of grade 2 or higher thrombocytopenia. The incidence of grade 3 (five out of 119, 4.2%) and grade 4 (two out of 119, 1.7%) thrombocytopenia was low and no bleeding episode was reported.

All patients underwent contrast-enhanced abdominal CT after oxaliplatin-based chemotherapy. The median time interval between the start of chemotherapy and the first follow-up CT scan was 49 days (interquartile range, 43–65 days), ranging from 34 to 90 days. The median spleen volume at follow-up was 215.94 cm3 (interquartile range, 170.65–273.89 cm3).

Radiomics signature construction and validation

After feature selection, LASSO algorithm identified five radiomics features with non-zero coefficients, including wavelet-LHL_firstorder_Energy, wavelet-HHL_GLCM_Idn, wavelet-HHL_GLSZM_HighGrayLevelZoneEmphasis, wavelet-LLL_firstorder_Kurtosis and wavelet-LLL_NGTDM_Complexity, which were strongly associated with the development of grade 2 or higher thrombocytopenia in the training cohort (Fig. 2). Rad-score was calculated for each patient using a linear combination of the 5 identified features multiplied by their respective coefficients. Calculation formula is presented in Supplementary Material. A significant difference in Rad-score was found between the TCP Group and the Non-TCP Group in the training cohort (−0.697 ± 0.357 vs −1.227 ± 0.325, p < 0.001) and further verified in the validation cohort (−0.900 ± 0.456 vs −1.288 ± 0.290, p =0.006). Patients in the TCP Group had significantly higher Rad-scores. The ROC curves indicated good performance of Rad-score in predicting grade 2 or higher thrombocytopenia in both cohorts (Fig. 3). The average AUC in the whole cohort from repeated 10-fold cross validation was 0.839 ± 0.146. Using the optimal cut-off value determined by maximizing the Youden’s index in the training cohort, Rad-score demonstrated favorable sensitivity, specificity and accuracy in the training cohort (81.8%, 84.1%, 83.5%, respectively) as well as the validation cohort (77.8%, 80.0%, 79.4%, respectively).

Figure 2 The least absolute shrinkage and selection operator (LASSO) regression algorithm for radiomics feature selection and radiomics signature construction.

(A) The penalization coefficient λ in the LASSO model was tuned using 10-fold cross-validation via minimum criteria. The binomial deviance was plotted against log(λ). The top x-axis is the number of variables for the given log(λ). The dotted vertical lines were drawn at the optimal values using the minimum criteria and the one-standard-error (1-SE) of the minimum. The optimal λ value of 0.108 was chosen. (B) LASSO coefficient profiles of the radiomic features. The LASSO coefficients were plotted against log(λ). Five features with non-zero coefficients were selected to construct radiomics signature based on the optimal λ value.

Figure 3 The receiver operating characteristic (ROC) curves shows good performance of the radiomics signature in predicting grade 2 or higher thrombocytopenia in both training and validation cohorts.

AUC: area under the curve. Numbers in parentheses are the 95% confidence interval.

Clinical model and clinical-radiomics model development and performance evaluation

Univariable logistic regression revealed that the cumulative dose of oxaliplatin, baseline BMI, liver metastases and spleen volume at follow-up were significantly associated with the development of grade 2 or higher thrombocytopenia. Patients who received higher cumulative dose of oxaliplatin, with liver metastases, lower baseline BMI and greater spleen volume at follow-up were significantly more likely to develop grade 2 or higher thrombocytopenia during chemotherapy. By contrast, age, gender, primary tumor site, chemotherapy regimen, number of chemotherapy cycles and bevacizumab treatment failed to show significant impacts on the risk of developing grade 2 or higher thrombocytopenia (Table 2).

Table 2 Univariable and multivariable logistic regression of clinical and radiomics variables for predicting grade 2 or higher thrombocytopenia during oxaliplatin-based chemotherapy.

Variables	Univariable analysis	Multivariable analysis	
			Clinical model	Clinical-radiomics model	
	OR (95% CI)	p value	OR (95% CI)	p value	OR (95% CI)	p value	
Age	0.988 (0.941,1.037)	0.619					
Gender		0.215					
Male	1						
Female	1.864 (0.697,4.982)						
BMI (kg/m2)	0.809 (0.659,0.993)	0.043	0.774 (0.607,0.986)	0.038			
Primary tumor site		0.196					
Colorectal cancer	1						
Gastric cancer	2.092 (0.683,6.408)						
Liver metastases		0.045					
No	1						
Yes	2.942 (1.022,8.469)						
Chemotherapy regimen		0.303					
CAPOX	1						
mFOLFOX6	0.483 (0.155,1.508)	0.210					
SOX	0.304 (0.034,2.680)	0.283					
Cumulative dose of oxaliplatin (mg/m2)	1.006 (1.002,1.009)	0.001	1.006 (1.002,1.009)	0.004	1.005 (1.001,1.009)	0.011	
Number of chemotherapy cycles	1.210 (0.982,1.490)	0.073					
Bevacizumab		0.653					
No	1						
Yes	0.688 (0.134,3.515)						
Spleen volume (cm3)	1.008 (1.001,1.015)	0.023	1.010 (1.002,1.018)	0.012			
Rad-Score	108.761 (12.640,935.862)	<0.001			124.190 (11.019,1399.701)	<0.001	

A clinical predictive model and a clinical-radiomics predictive model were constructed using the statistically significant clinical factors in the univariable analysis alone and together with radiomics signature respectively (Table 2). Baseline BMI, cumulative dose of oxaliplatin and spleen volume were the 3 factors included in the clinical model. Rad-score and cumulative dose of oxaliplatin were the only 2 factors in the clinical-radiomics model. The clinical-radiomics model was presented as a nomogram in Fig. 4. As shown in Fig. 5 and Table 3, compared to the clinical model, the clinical-radiomics model manifested higher sensitivity, specificity and accuracy in discriminating patients who developed grade 2 or higher thrombocytopenia from those who did not in both training and validation cohorts. In the training cohort, the AUC of the clinical-radiomics model and clinical model was 0.913 (95% CI [0.851–0.974]) and 0.828 (95% CI [0.720–0.935]), respectively. In the validation cohort, the AUC of the clinical-radiomics model and clinical model was 0.867 (95% CI [0.727–1.000]) and 0.751 (95% CI [0.564–0.938]), respectively. In the whole cohort, the average AUC from repeated 10-fold cross validation was higher for the clinical-radiomics model compared with the clinical model (0.882 ± 0.106 vs 0.784 ± 0.154) though not statistically significant (bootstrap method, p = 0.057). However, IDI index showed that integrating radiomics signature to conventional clinical factors significantly improved the predictive accuracy by 17.0% (95% CI [4.9%–29.1%], p = 0.006) in the whole cohort. The Hosmer-Lemeshow goodness-of-fit test yielded non-significant p values (clinical model, p = 0.537 (training cohort) and 0.369 (validation cohort), and clinical-radiomics model, p = 0.893 (training cohort) and 0.257 (validation cohort)), suggesting favorable calibration of both models. However, the AIC value of the clinical-radiomics model was substantially lower than that of the clinical model (62.316 vs 81.789, ΔAIC = 19.473), suggesting a significantly better calibration of the clinical-radiomics model. The calibration curves of the two models in the training and validation cohorts were showed in Fig. 6. Figure 7 showed the clinical-radiomics nomograms of two cases for predicting the risk of developing grade 2 or higher thrombocytopenia during oxaliplatin-based chemotherapy.

Figure 4 The established clinical-radiomics nomogram for predicting the risk of grade 2 or higher thrombocytopenia during oxaliplatin-based chemotherapy.

TCP: thrombocytopenia.

Figure 5 The receiver operating characteristic (ROC) curves of the clinical model and the clinical-radiomics model in discriminating grade 2 or higher thrombocytopenia.

In the training cohort, the area under the curve (AUC) of the clinical model and the clinical-radiomics model was 0.828 (95% CI [0.720–0.935]) and 0.913 (95% CI [0.851−0.974]), respectively. In the validation cohort, the AUC of the clinical model and the clinical-radiomics model was 0.751 (95% CI [0.564–0.938]) and 0.867 (95% CI [0.727−1.000]), respectively.

Table 3 Diagnostic performance of the clinical model and the clinical-radiomics model in the training and validation cohort.

	Training cohort		Validation cohort	
	Clinical model	Clinical-radiomics model		Clinical model	Clinical-radiomics model	
AUC
(95% CI)	0.828
(0.720–0.935)	0.913
(0.851–0.974)		0.751
(0.564–0.938)	0.867
(0.727–1.000)	
Sensitivity	77.3%	86.4%		66.7%	66.7%	
Specificity	82.5%	84.1%		80.0%	92.0%	
Accuracy	81.2%	84.7%		76.5%	85.3%	
Notes.

AUC area under the curve

Figure 6 Calibration curves of the clinical model and the clinical-radiomics model in the training cohort (A) and validation cohort (B).

Figure 7 Clinical-radiomics nomograms of 2 cases for predicting the risk of grade 2 or higher thrombocytopenia during oxaliplatin-based chemotherapy.

(A) A 60-year-old male gastric cancer patient received CAPOX regimen. Cumulative dose of oxaliplatin was 756 mg/m2, Rad-score was −0.297 and the total point of nomogram was 114.9. This patient developed grade 3 thrombocytopenia during chemotherapy; (B) A 50-year-old female gastric cancer patient received CAPOX regimen. Cumulative dose of oxaliplatin was 724 mg/m2, Rad-score was −1.263 and the total point of nomogram was 64.9. This patient did not develop thrombocytopenia during chemotherapy.

Discussion

In this study, a whole-spleen radiomics signature based on follow-up contrast-enhanced portal venous phase CT images was developed and validated for the prediction of grade 2 or higher thrombocytopenia in patients with gastrointestinal malignancies treated with oxaliplatin-based chemotherapy. This radiomics signature manifested good performance in risk prediction of thrombocytopenia and provided incremental values over conventional clinical factors as the clinical-radiomics predictive model, incorporating the radiomics signature with clinical factors outperformed the clinical predictive model in both discrimination and calibration.

Although oxaliplatin-induced severe or life-threatening (grade 3 or 4) thrombocytopenia is relatively rare (Grothey et al., 2018), which is consistent with our study, even moderate (grade 2) thrombocytopenia can adversely affect chemotherapy dose intensity, since for persistent grade 2 or higher thrombocytopenia which does not recover to grade 0 or 1 by the start of next cycle, discontinuation, delay initiation or dose reduction of chemotherapy are required. Early prediction of grade 2 or higher thrombocytopenia is critical to ensure close monitoring of platelet count and bleeding risk, and early management to minimize the adverse effects. For example, in patients at high risk, prophylactic medication of recombinant human thrombopoietin (rh-TPO) or thrombopoietin receptor agonists (TPO-RA) might be administered to prevent severe thrombocytopenia and to maintain chemotherapy dose intensity (Kuter, 2022; Jin et al., 2021; Wang et al., 2018). However, there is no well-established method to early identify patients at high risk of oxaliplatin-related thrombocytopenia. In our study, the whole-spleen radiomics signature based on follow-up CT images acquired at a median time interval of 49 days after the start of oxaliplatin-based chemotherapy could predict grade 2 or higher thrombocytopenia which developed at a median time interval of 109 days with considerable accuracy.

Previous studies have demonstrated that one of the important mechanisms of oxaliplatin-related thrombocytopenia might be hepatic sinusoidal injury, also known as sinusoidal obstruction syndrome (SOS) (Jardim et al., 2012; Overman et al., 2010). Oxaliplatin can damage hepatic sinusoidal endothelial cells leading to disruption of sinusoidal barrier, followed by collagen deposition in perisinusoidal space, stenosis and occlusion of hepatic venules, portal hypertension and eventually splenomegaly and splenic sequestration with associated thrombocytopenia (Rubbia-Brandt et al., 2004; Aloia et al., 2006; Rubbia-Brandt et al., 2010).

Some researchers have found a strong correlation between splenomegaly and oxaliplatin-related thrombocytopenia (Overman et al., 2010; Kim et al., 2016; Miyata et al., 2020; Jung et al., 2012). Similar to their findings, spleen volume at follow-up was an independent factor for thrombocytopenia in our clinical predictive model. In addition, our whole-spleen radiomics signature demonstrated a significant association with oxaliplatin-related thrombocytopenia and outperformed spleen volume in risk prediction. In the present study, radiomics signature was developed based on abdominal contrast-enhanced CT, the standard of care for the imaging surveillance in gastrointestinal malignancies nowadays. With the readily available open-source software, the whole-spleen could be semi-automatically segmented and radiomics features could be extracted within a few minutes. Radiomics converts digital medical images into minable high-dimensional data, having a potential to uncover disease characteristics and provide non-invasive pathophysiological information complementary to clinical data (Gillies, Kinahan & Hricak, 2016), thus might act as biomarkers for early prediction of disease progression and prognosis in a variety of neoplastic and non-neoplastic diseases (Dohan et al., 2020; Park et al., 2020; Ji et al., 2020; Lin et al., 2020; Li et al., 2021; Martini et al., 2021). Theoretically, our contrast-enhanced CT-based whole-spleen radiomics signature could reflect the pathophysiological changes of oxaliplatin-induced splenomegaly and splenic sequestration, hence could serve as a biomarker for the associated thrombocytopenia. Our radiomics signature was composed of 5 radiomics features obtained after wavelet transform, which is a process to suppress noise and highlight details in the original CT images. The first-order features firstorder_Energy and firstorder_Kurtosis measure the magnitude and distribution of voxel values in the image, respectively. The feature firstorder_Energy is, to some extent, positively correlated to the volume. On the other hand, the second- and higher-order features (GLCM_Idn, GLSZM_HighGrayLevelZoneEmphasis and NGTDM_Complexity) measure the spatial arrangement of voxel values and local heterogeneity (Zwanenburg et al., 2019; Ak et al., 2022). Although it is still challenging to interpret the exact biological meanings of the radiomics features, we hypothesized that oxaliplatin-induced splenomegaly and splenic sequestration might lead to an increase in splenic volume and local heterogeneity, thus lead to an increase in the radiomics features and Rad-score. Therefore, patients who developed grade 2 or higher thrombocytopenia had significantly higher Rad-score.

The cumulative dose of oxaliplatin was an independent predictor for thrombocytopenia in our clinical model as well as the clinical-radiomics model. Similar to our findings, studies showed that oxaliplatin-based chemotherapy resulted in a dose-dependent splenomegaly, which was correlated with thrombocytopenia (Overman et al., 2010; Kim et al., 2016). BMI, an easily measured indictor for the assessment of nutritional status, was another independent clinical factor for predicting thrombocytopenia in our clinical model. Low BMI was significantly associated with hematological adverse events, such as neutropenia and anemia (Razzaghdoust, Mofid & Moghadam, 2018; Razzaghdoust, Mofid & Peyghambarlou, 2020) during chemotherapy in previous studies. Similarly, low BMI was an independent clinical predictor for chemotherapy-induced thrombocytopenia in patients with solid tumors and lymphoma (Razzaghdoust, Mofid & Zangeneh, 2020), which was in accordance with our findings, despite the fact that we focused on oxaliplatin-based chemotherapy in gastrointestinal malignancies.

Brouquet et al. (2009) found that synchronous liver metastases might be an independent risk factor for hepatic sinusoidal injury after chemotherapy, however, they did not report platelet counts in their article. Although our univariable analysis discovered that patients with liver metastases at diagnosis were more likely to develop grade 2 or higher thrombocytopenia during oxaliplatin-based chemotherapy, this trend was not confirmed in the multivariable analysis. Previous evidences suggested that concurrent bevacizumab treatment may protect against oxaliplatin-related SOS, with resultant reduction in splenomegaly and thrombocytopenia (Overman et al., 2010; Overman et al., 2018; Allegra et al., 2009; Hubert et al., 2013). However, our present study failed to reveal this protective effect of bevacizumab, which may be explained by the small proportion of patients receiving bevacizumab in our cohort. Future studies are warranted to investigate the role of synchronous liver metastases and bevacizumab in oxaliplatin-related thrombocytopenia.

There are several limitations to our study. First, owing to the retrospective nature of this study, there might be selection bias and confounding factors. Second, the radiomics signature and clinical-radiomics model were developed based on our single-center data with relatively small sample size. Therefore, future prospective studies with larger sample size are needed to further validate our results. Third, with the advances in the understanding of the mechanisms of oxaliplatin-related thrombocytopenia, the predictive values of synchronous liver metastases, bevacizumab treatment and other emerging potential clinical factors require further investigation to improve our clinical-radiomics model.

Conclusions

In conclusion, contrast-enhanced CT-based whole-spleen radiomics signature might serve as an early predictor for grade 2 or higher thrombocytopenia during oxaliplatin-based chemotherapy in patients with gastrointestinal malignancies and provide incremental value over conventional clinical factors.

Supplemental Information

Supplemental Information 1 An example of the semi-automatic segmentation of the whole spleen using the Nvidia AI-Assisted Annotation (Nvidia AIAA) extension implemented in 3D Slicer

(A) Boundary points were set near the edge of spleen on axial, sagittal and coronal views. (B) Semi-automatic segmentation of 3D Slicer software. (C) Interactive correction of the segmentation if necessary. Large blood vessels were carefully excluded. (D) The final segmentation.

Click here for additional data file.

Supplemental Information 2 Oxaliplatin-based chemotherapy regimens

Click here for additional data file.

Supplemental Information 3 Rad-score calculation formula

Click here for additional data file.

Supplemental Information 4 Raw data

Click here for additional data file.

Additional Information and Declarations

Competing Interests

Author Contributions

Human Ethics

Data Availability

The authors declare there are no competing interests.

Yuhong Dai performed the experiments, analyzed the data, prepared figures and/or tables, authored or reviewed drafts of the article, and approved the final draft.

Yiqi Cheng performed the experiments, analyzed the data, prepared figures and/or tables, authored or reviewed drafts of the article, and approved the final draft.

Ziling Zhou performed the experiments, authored or reviewed drafts of the article, and approved the final draft.

Zhen Li conceived and designed the experiments, analyzed the data, authored or reviewed drafts of the article, and approved the final draft.

Yan Luo analyzed the data, authored or reviewed drafts of the article, and approved the final draft.

Hong Qiu conceived and designed the experiments, authored or reviewed drafts of the article, and approved the final draft.

The following information was supplied relating to ethical approvals (i.e., approving body and any reference numbers):

This retrospective study was approved by the Ethics Committee of Tongji Hospital, Tongji Medical College, Huazhong University of Science and Technology.

The following information was supplied regarding data availability:

The raw data are available in the Supplemental Files.

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
