# Peer review of "A contrast-enhanced CT-based whole-spleen radiomics signature for early prediction of oxaliplatin-related thrombocytopenia in patients with gastrointestinal malignancies: a retrospective study"

_PeerJ, doi:10.7717/peerj.16230_

## Round 0.1 · original submission · Minor Revisions

Thank you very much for submitting this important study to PeerJ. Overall, the reviewers had a very favorable review regarding your manuscript. However, several points were raised. Most importantly, please add details regarding the model construction to improve replicability.

Reviewer 1 ·

Basic reporting

Reviewer Comments:

1. The work presented in the manuscript is both interesting and highly valuable in the medical field. The research conducted by the authors has the potential to contribute significantly to the advancement of medical knowledge and practice.

2. The authors have done an excellent job of presenting their findings in a clear, unambiguous, and technically accurate manner. The manuscript is well-structured, and the information provided is concise and easy to comprehend. This approach enhances the overall readability of the article and ensures that the key points are effectively communicated.

3. The technical accuracy of the manuscript is commendable.

Experimental design

1. The results presented in this study are compelling and persuasive. The data and findings provided offer substantial support to the authors' claims and conclusions.

2. However, I strongly suggest that the authors provide additional details about the models utilized for analysis, rather than solely mentioning the software packages employed. By including specific information about the models, such as their architecture, parameters, and any modifications made, the authors can enhance the clarity and transparency of their research methodology.

3. It is crucial for the readers to have a comprehensive understanding of the authors' entire body of research. Therefore, by incorporating the aforementioned model details, the authors can facilitate the readers' comprehension and enable them to reproduce or build upon the study in a more informed manner.

Validity of the findings

Overall, I find the presented results to be convincing. However, I encourage the authors to address this specific concern by including comprehensive information about the models used, thereby strengthening the research and facilitating a better understanding of the study for the readers.

·

Basic reporting

no comments

Experimental design

Study is appropriately designed
Method is valid and reliable
It might be worth expanding in line 35 what percentage had thrombocytopenia in those 119 patients and between 37-38 to give timeline abstract as well
Please use full form in line 102 and line 116

Validity of the findings

subject selection is appropriate and variables well defined. Authors presented great detail in order to replicate the study
Conclusion answers the aim of study
Conclusion is supported by results and well correlated with existing evidence via references
Limitation are well described which brings opportunity for future research

Additional comments

1. It is a complication which needs attention.
2. Research question is clearly outlined
3. it is well justified to explore in this field
4. Study is appropriately designed
5. Method is valid and reliable
6. It might be worth expanding in line 35 what percentage had thrombocytopenia in those 119 patients and between 37-38 to give timeline abstract as well
7. Please use full form in line 102 and line 116
8. subject selection is appropriate and variables well defined. Authors presented great detail in order to replicate the study
9. Results are clearly stated
10. Data is presented in clear and appropriate manned for international readers
11. Results are discussed nicely to correlate with existing evidence and how it fits with their study. Between 281 and 284, it might be good idea for authors to elaborate on how early diagnosis/prediction can alter the course of treatment by giving evidence of what pathway treatment must change to improve survival/treatment
12. Conclusion answers the aim of study
13. Conclusion is supported by results and well correlated with existing evidence via references
14. Limitation are well described which brings opportunity for future research

---

## Round 0.2 · accepted · Accept

Thank you very much for submitting your work to the journal. The reviewers are satisfied with the revisions.

Reviewer 1 ·

Basic reporting

The authors have sincerely addressed all my suggestions. The manuscript in its current form may be accepted.

Experimental design

Acceptable

Validity of the findings

Acceptable

Additional comments

Nil

·

Basic reporting

no comments

Experimental design

Study is appropriately designed
Method is valid and reliable
It might be worth expanding in line 35 what percentage had thrombocytopenia in those 119 patients and between 37-38 to give timeline abstract as well
Please use full form in line 102 and line 116- I appreciate authors providing the asked insight

Validity of the findings

subject selection is appropriate and variables well defined. Authors presented detail to replicate the study
Conclusion answers the aim of study
Conclusion is supported by results and well correlated with existing evidence via references
Limitation are well described which brings opportunity for future research

Additional comments

1. It is a complication which needs attention.
2. Research question is clearly outlined
3. It is well justified to explore this field
4. Study is appropriately designed
5. Method is valid and reliable
6. It might be worth expanding in line 35 what percentage had thrombocytopenia in those 119 patients and between 37-38 to give timeline abstract as well- Thank you
7. Please use the full form in line 102 and line 116- thank you
8. subject selection is appropriate and variables well defined. Authors presented great detail in order to replicate the study
9. Results are clearly stated
10. Data is presented in a clear and appropriate manner for international readers
11. Results are discussed nicely to correlate with existing evidence and how it fits with their study. Between 281 and 284, it might be good idea for authors to elaborate on how early diagnosis/prediction can alter the course of treatment by giving evidence of what pathway treatment must change to improve survival/treatment- I appreciate authors making the addition
12. Conclusion answers the aim of study
13. Conclusion is supported by results and well correlated with existing evidence via references
14. Limitation are well described which brings opportunity for future research